# Mindfulness-Based Cognitive Therapy for Stress Reduction in Family Carers of People Living with Dementia: A Systematic Review

**DOI:** 10.3390/ijerph19010614

**Published:** 2022-01-05

**Authors:** Emme Chacko, Benjamin Ling, Nadav Avny, Yoram Barak, Sarah Cullum, Fred Sundram, Gary Cheung

**Affiliations:** 1Faculty of Medical and Health Sciences, The University of Auckland, Auckland 1142, New Zealand; benjamin.ling@auckland.ac.nz (B.L.); s.cullum@auckland.ac.nz (S.C.); f.sundram@auckland.ac.nz (F.S.); gary.cheung@auckland.ac.nz (G.C.); 2Auckland District Health Board, Auckland 1023, New Zealand; NadavA@adhb.govt.nz; 3Department of Psychological Medicine, Dunedin School of Medicine, University of Otago, Dunedin 9016, New Zealand; Yoram.barak@otago.ac.nz

**Keywords:** mindfulness, carers, dementia, stress, cognitive therapy, depression, systematic review

## Abstract

The prevalence of dementia is increasing and the care needs of people living with dementia are rising. Family carers of people living with dementia are a high-risk group for psychological and physical health comorbidities. Mindfulness-based interventions such as mindfulness-based cognitive therapy show potential for reducing stress experienced by family carers of people living with dementia. This study aims to systematically assess the efficacy of mindfulness-based cognitive therapy in reducing stress experienced by family carers of people living with dementia. Electronic databases including MEDLINE, APA PsycINFO, EMBASE, CINAHL, Scopus, Web of Science, Cochrane Library, AMED, ICTRP, and ALOIS were searched for relevant studies up to August 2020. All types of intervention studies were included. Quantitative findings were explored. Seven studies were eligible for inclusion. The analysis showed that there was a statistically significant reduction in self-rated carer stress in four studies for the mindfulness-based cognitive therapy group compared to controls. One study that was adequately powered also showed reductions in carer burden, depression, and anxiety compared to control. Mindfulness-based cognitive therapy appears to be a potentially effective intervention for family carers of people living with dementia, but large, high-quality randomized controlled trials in ethnically diverse populations are required to evaluate its effectiveness.

## 1. Introduction

As the world’s population grows and ages, the prevalence of dementia is rising rapidly [1]. Dementia is associated with a decline in cognitive function and the inability to perform activities of daily living, which results in substantial ongoing care needs for people with dementia as the disease progresses.

Family carers provide the majority of the care involved for people living with dementia (PLWD) [2] in a largely unpaid manner; therefore, saving society considerable costs of this care. It is well known that being a dementia carer is a risk factor for psychological stress [3] and poor physical health [4]. Approximately 40% of these carers experience clinical depression or anxiety [5]. Of particular concern are findings from a UK survey of 566 dementia carers which showed that 16% were suicidal [6].

### 1.1. Psychological Factors Related to Carers of PLWD

The experience of carer stress is significantly increased by the presence of behavioural and psychological symptoms of dementia (BPSD) in PLWD [7,8] such as agitation, apathy, wandering, and psychosis. Higher levels of stress may be experienced by carers looking after older adults with physical disabilities [9]. Carer stress is also affected by factors such as the severity of the cognitive decline in PLWD [10], the duration of caregiving [11], being older, female, and living with PLWD [7,8].

The quality of dyadic relationships between family carers and PLWD is important because closer relationships are predictive of positive outcomes for both the carers [12] and PLWD [13,14]. Without intervention, carer stress could increase the likelihood of premature entry into aged residential care [15] and elder abuse [16]. Therefore, there is an urgent clinical need and increasingly economic argument to provide dementia carers with cost-effective and sustainable stress-reduction interventions.

### 1.2. Traditional Stress-Reduction Interventions

Many psychosocial stress-reduction interventions such as respite, educational workshops, skills training, and support groups are offered to carers of PLWD. However, systematic reviews have shown that the effect on stress reduction from these interventions is not significant [17]. The evidence supporting psychological interventions for stress reduction in family carers is inconsistent and weak [10], transient when present [18], and lacking in specificity [3]. Interventions that require active participation are associated with greatest effect [19]. A recent meta-analysis of high quality but significantly heterogenous study designs showed that psychosocial interventions have a small to moderate effect on dementia carer burden, depression, and general health [20], but not an overall effect on quality of life (QOL).

Cognitive Behavioural Therapy (CBT) is the most widely studied psychotherapy that is used for depressed dementia carers [21]. However, despite a substantial evidence base for depression, concerns are increasing about the effect sizes of CBT being relatively small, with sizes in the range of 0.10–0.36 for carers [21] and effects not enduring over time [22,23]. Therefore, there is a need to explore other cost-effective interventions of more enduring benefit.

### 1.3. Mindfulness-Based Interventions

Mind-body interventions such as mindfulness are increasing in popularity, and there is some evidence for their use with family dementia carers. The most well-known mindfulness-based interventions (MBI) are the Mindfulness-Based Stress Reduction (MBSR) program [24] and Mindfulness-Based Cognitive Therapy (MBCT) [25]. MBSR was originally developed for patients with chronic pain. MBCT was based on MBSR, but with cognitive behavioural techniques added to the MBSR-style practices [25]. MBCT’s original indication was for recurrent depression, where it was shown to be effective in the prevention of relapse in a meta-analysis of six RCTs in various countries involving 593 participants [26]. Importantly, MBCT has been shown to be as effective as antidepressant medication treatment for the prevention of relapse into depression, and may be more effective than medication in those with histories of severe childhood abuse [27]. MBCT is recommended for recurrent depression in clinical practice guidelines both in the UK since 2004 [28] as well as Australia and New Zealand since 2015 [29]. Since its original use for depression, it has been applied to many other indications with good effect [30].

MBIs have been used with family carers of other chronic diseases such as cancer [31] and developmental disabilities [32]. Preliminary evidence from a recent systematic review suggests MBIs (mostly MBSR) are effective for stress reduction in family dementia carers [33]. A meta-analysis was performed with three (144 participants) of the five pilot studies included in this systematic review [34,35,36] and showed a significant reduction in stress levels after the MBI, with a moderate aggregated effect size of 0.57 (95% CI [0.23–0.92]).

### 1.4. Research Gap

A systematic review looking specifically at the efficacy of MBCT on outcomes in dementia carers has not been conducted to date, and there appear to be a number of trials in this area that need to be summarized. MBCT has a central principle of encouraging approach toward negative experiences rather than reacting with aversion. This principle shows particular promise in this population of carers because of the evidence from a systematic review that denial, avoidance, and wishful thinking as coping strategies are associated with poor outcomes for dementia carers [3]. Additionally, given that MBCT targets depressive rumination specifically [25] it holds more potential over other MBIs in the carer populations where there are high rates of clinical depression and anxiety. This justifies the need to conduct a systematic review looking specifically at its efficacy in this population separate to other MBIs.

### 1.5. Aims

The primary aim of this systematic review was to examine the evidence for MBCT to reduce carer stress in family carers of PLWD when compared with treatment as usual, waitlist, or no control. Secondary aims are to review the evidence for MBCT to reduce carer burden, depression, and to increase QOL, resilience, and wellbeing. Other secondary aims are to review the evidence of MBCT to improve BPSD in people being cared for, and whether there are any reports of harms associated with the use of MBCT in this population. We will also review whether MBCT changes measures of trait mindfulness in carers.

## 2. Materials and Methods

This systematic review was registered on PROSPERO (CRD42020186414) on 5 May 2020 and the PRISMA 2009 [37] reporting checklist was used.

### 2.1. Search Strategy

Systematic searches were conducted between 1 June 2020 and 1 August 2020 using the following electronic databases: MEDLINE (via OVID), APA PsycINFO (via OVID), EMBASE (via OVID), CINAHL (via EBSCOhost), Scopus (via ELSEVIER), Web of Science, Cochrane Library (Wiley Interface), AMED, ICTRP, and ALOIS. Unpublished literature was also searched in ProQuest Dissertations and Theses Global, Google Scholar and MedNar. Search alerts were enabled in all databases to ensure ongoing retrieval of relevant studies. A hand search of reference lists of all relevant articles identified and of the *Mindfulness* journal was performed. Experts were contacted to ensure saturation of literature (authors of four identified studies were emailed to ask about other studies that they were aware of). Keywords used included “Mindful*”, “MBCT”, “Dementia”, “Alzheimer”, “Cognit*”, “neurocognit*”, “care*”, “caregive*” and other relevant subject headings of each database. A full search strategy for MEDLINE (via OVID) is available in the Appendix A.

### 2.2. Eligibility Criteria

#### 2.2.1. Study Types

Any experimental study design was included, such as randomized controlled trials (RCTs), quasi-experimental, prospective, or retrospective cohort studies that evaluated the efficacy of MBCT in family carers of PLWD. Studies with any type of control (treatment as usual, active, or inactive controls) were included. Studies of all languages were attempted to be included as long as translation resources were available. There was no restriction on setting and study duration. Studies were included from unpublished sources if data were available.

#### 2.2.2. Participant Types

Studies involving family carers of any age were included. Family carers were defined as spouses, children, grandchildren, siblings, other relatives of a PLWD or person with significant cognitive impairment. Carers did not have to be blood relatives. Staff and paid caregivers were excluded.

#### 2.2.3. Intervention Types

MBCT or adaptations of MBCT were included. Both online and in-person group formats of MBCT were included.

#### 2.2.4. Outcome Measurement Types

The primary outcome of this systematic review was carers’ perceived stress levels. This was chosen as this appears to be the main way to assess efficacy of MBCT interventions in a manner that is relevant to carers. Secondary outcomes were carer burden, depression, QOL, resilience, wellbeing, trait mindfulness, BPSD in the PLWD, and potential adverse effects. The secondary outcomes were chosen because mindfulness interventions can improve a raft of other health outcomes [24], with potential for benefit on dyadic interactions involving PLWD. Trait mindfulness was also chosen because it is the process indicator that explains the change in other outcomes.

### 2.3. Data Extraction

Literature search results were transferred to reference management software (RefWorks). COVIDENCE, a systematic review software for screening and data extraction, was used. There was a first pass extraction using titles and abstracts from studies retrieved using the search strategy. This was conducted independently by two blinded review authors (EC, NA) to identify studies that potentially met the inclusion criteria outlined above. The second pass extraction involved retrieving the full text of the studies and independent assessment for eligibility by two blinded review authors (EC, NA). Disagreement between reviewers was resolved through a third review author (GC) through discussion. A log of excluded studies was kept with reasons at the full text screening stage. Two blinded reviewers extracted data independently (EC, BL) and discrepancies were identified and resolved through discussion with a third author (GC) where necessary.

Missing data were requested from study authors. Extracted information included setting, year, design, sociodemographic characteristics of the carers, intervention and control group details, outcomes, and suggested improvements.

### 2.4. Risk of Bias

We used Cochrane Collaboration’s revised risk of bias tool (RoB 2) to assess the risk of bias in the RCTs [38]. RoB 2 was used by two blinded review authors (EC, YB) who independently assessed the risk of bias in all RCTs. RoB 2 is structured into five domains to assess biases in the following areas: randomization process, deviations from intended interventions, missing outcome data, measurement of the outcome, and selection of the reported result. Assessment results were discussed between the two blinded review authors and taken to a third author (GC) for a final decision made by consensus. The aim of this assessment was to determine the quality of the evidence presented by the studies, but all studies were still included regardless of their risk of bias.

### 2.5. Data Analysis

It was not anticipated that meta-analyses would be conducted due to heterogeneity of studies. Instead, a systematic review approach was planned with information presented in text and tables to summarize and explain the characteristics and outcomes of included studies. Findings both within and between included studies were explored. Findings were presented in order of main and additional outcomes.

## 3. Results

The search strategy including unpublished grey literature resulted in 595 articles (Figure 1). After duplicate removal there were 318 results. The first pass screening using titles and abstracts removed protocols, studies of non-MBCT interventions, and non-dementia carer populations. The remaining 18 results were screened using full texts. Two studies were not in English and translators screened these. A total of 11 results were excluded for the following reasons: five wrong interventions, one wrong patient population, three duplicates, one was not an intervention study, and one study could not be retrieved despite extensive searches using inter-library services and contacting the author, journal, and publisher. Ultimately, seven studies with a total of 291 subjects that fulfilled selection criteria were included for analysis.

### 3.1. Study Characteristics

#### 3.1.1. Study Design and Control Group Conditions

The study designs of all studies are presented in Table 1. The seven studies included were those by Cheung et al., (2020) [39]; Kor et al., (2019, 2020) [40,41]; Norouzi et al., (2015) [42]; Oken et al., (2010) [36]; Ozen (2013) [43]; and Zarei (2018) [44]. All included studies were randomized and controlled. Three studies were described as pilot RCTs [35,40,43]. The number of participants in the seven studies ranged from 12–113, with only two studies [39,41] recruiting over 50 participants.

Oken et al., (2010) [35] was a three-armed RCT where participants were divided into an MBCT intervention group and two control groups (education group as active control and respite group as pragmatic control). Three other studies used active control groups [38,39,40]. Two studies [40,41] used brief education programs as the control that matched the MBCT group in terms of duration and number of sessions. These education sessions were also structured to provide a mix of didactic teaching and group sharing experiences which is similar to MBCT. The Cheung et al., (2020) study [39] compared MBCT with MBSR. MBCT and MBSR are similar in terms of duration and number of sessions. The other studies [42,43,44] used inactive control groups of usual care or waitlist. The Ozen (2013) [43] study included dyads of carers and PLWD, or carers alone, and was performed as an unblinded crossover RCT.

#### 3.1.2. Family Carer Characteristics

The characteristics of participants in all studies are presented in Table 1. In all seven studies, most of the participants were women (ranging from 61–100%). The mean age ranged from 57.1–68.9 years. Whilst the studies were conducted in a range of countries (USA, Canada, Hong Kong, Iran), formal ethnicity data were only reported in two studies. Oken et al. (2010) [35] had mostly Caucasian participants (90.3%), with small numbers of African American (3.2%), and Asian (6.5%) participants. Participants in the Cheung et al. (2020) [38] study were all Chinese.

The carer’s relationship with PLWD was quite variable in the seven studies and included spouses (7.5–100%) as well as children. The mean duration of caregiving which was reported in four studies [39,40,41,44] ranged from 5.1–8.7 years.

#### 3.1.3. Intervention

A summary of interventions used in all studies is presented in Table 1. In all studies, the experimental intervention was some form of MBCT. The original format of MBCT consists of eight, 2.5 h weekly sessions and a whole-day retreat. Almost all studies modified the MBCT protocol in some way from its original format. Only Norouzi et al. (2015) [42] did not provide details about any modifications to the MBCT protocol. Zarei (2018) [44] delivered the MBCT online as tele-MBCT, whilst all other studies used the original in-person group format. The Zarei (2018) study [44] also used the self-help book, *The Mindful Way Workbook* [45] to supplement the MBCT. The modifications were made by a panel of expert clinicians in three of the studies [39,40,41]. The other studies did not describe who made the modifications.

Adaptations were made to tailor the MCBT specifically for carers. In the Kor et al. (2020) [41] study, this included psychoeducation about stress and replacement of depression relapse content with dementia caregiving skills. Additionally, used were responding to negative moods associated with caregiving, and the identification of habitual emotional reactions to difficulties in caregiving. The Zarei (2018) [44] study also made adaptations to content to include issues of carer identity and ambiguous loss. This study modified the movement practices to enhance carer safety. The Oken et al. (2010) [35] study included a shared education session on dementia with the active control group. This [39] study described a focus on CBT concepts to help carers gain confidence early.

#### 3.1.4. Outcome Measures

The primary outcome of this systematic review was the carers’ perceived stress level. Secondary outcomes were carer burden, depression, QOL, resilience, wellbeing, trait mindfulness, BPSD in PLWD, and potential side effects.

The perceived stress of carers in most studies was measured using the Perceived Stress Scale (PSS) [46]. This was measured at pre- and post-MBCT intervention, as well as at three months post intervention in Kor et al., (2019) [40], and six months post intervention in Kor et al., (2020) [41] to see if effects were sustained. The Oken et al., (2010) [35] study used the Revised Memory and Behavior Problems Checklist [47] as their primary outcome measure, which has the stress reaction of carers as one of two main components. Oken et al., (2010) [35] also measured salivary cortisol and inflammatory markers (IL-6, TNF-alpha, CRP) as additional and more objective measures of carer stress.

Carer burden was measured using the Zarit Burden Interview (ZBI) [48] in three studies ([39,40,41], its shortened version [49] in Ozen (2013) [43] or the Caregiver Burden Inventory (CBI) [50] in Norouzi et al., (2015) [42].

Depression was measured using the Center for Epidemiological Studies–Depression Scale (CES-D) [51] in most studies [36,39,40,41,44]. The Ozen (2013) [43] study used the 30-item Geriatric Depression Scale [52] and the Depression Anxiety Stress Scale [53]. The Hamilton Depression Rating Scale [54] was used in Norouzi et al., (2015) [42].

Resilience was measured using the Brief Resilience Scale (BRS) [55] in the Kor et al. (2019, 2020) [40,41] studies.

QOL for carers was measured using Short From 12 Physical Component Summary Score (SF-PCS) [56] and the Short Form 12 Mental Component Summary Score (SF12-MCS) [56] in both those [40,41] studies.

BPSD in PLWD was measured using the Apathy Evaluation Scale—Informant Version (AES) [57] in Oken et al., (2010) [35] and the Neuropsychiatric Inventory—Questionnaire (NPI-Q) [58] in Kor et al., (2020) [41].

Trait mindfulness was measured using the Five-Facet Mindfulness Questionnaire (FFMQ) [59] in four studies [39,40,41,43]). The Oken et al., (2010) [35] study used the Mindful Attention Awareness Scale [60].

### 3.2. Risk of Bias Assessment of Included Studies

For all studies, the Cochrane Collaboration’s Risk of Bias 2 (RoB 2) tool [38] was used because all studies were randomized and controlled in design. The assessment of bias for all studies is reported in Figure 2.

#### 3.2.1. Randomization

The Ozen (2013) [43] and Norouzi et al., (2015) [42] studies scored a high risk of bias due to randomization processes. Neither study reported details of sequence generation, allocation concealment, and baseline characteristics. The Oken et al., (2010) [35] study also had the same high risk scores across those domains but did report an adequate randomization sequence process.

#### 3.2.2. Deviations

Four studies [35,42,43,44] scored some concerns in the domain of bias due to deviations because of lack of reported blinding and intention to treat protocols. Participant blinding is challenging for mindfulness interventions such as MBCT and none of the included studies were able to do this. However, three studies [39,40,41] scored low risk in the domain where this was assessed (bias due to deviations from intended interventions) as a result of the algorithm allowing for this to be compensated by other more favourable aspects of risk such as intention to treat procedures.

#### 3.2.3. Outcome Data

Four studies [35,42,43,44] scored high risk for missing outcome data. Three of these studies reported a significant attrition rate without reasons or appropriate statistical analysis to manage this, while the Norouzi et al., (2015) [42] study did not explicitly comment on attrition and was therefore also rated as high risk for this measure. The Kor et al., (2019) [40] study had some concerns in the domain of missing outcome data because two participants were lost to follow up with reasons that may have been significant in this small sample.

#### 3.2.4. Outcome Measures

All studies were deemed high risk by virtue of having participant reported self-rated scores as their main outcome measure. According to the RoB 2 tool, the outcome assessors are study participants if measures are self-reported [38]. Self-report introduces social desirability bias as it is likely that the assessment of outcome is influenced by knowledge of the received intervention. This was the case for all included studies as no participants could be blinded to the intervention.

However, the Oken et al., (2010) [35] study also used objective physiological markers measured by blinded outcome assessors, and these outcomes would have had a low risk of bias in outcome assessment. Oken et al. [35] reported that participant “expectancies” had been assessed to be the same between groups, but no further details were available. However, since they also included subjective assessments, the overall risk remained high.

#### 3.2.5. Selection of Results

Four studies [35,42,43,44] also scored some concerns in the domain of bias due to selection of results because there was no study protocol available.

#### 3.2.6. Overall Risk of Bias

Setting the issue of outcome measurement aside, the studies then ranged in their risk of bias with some studies scoring well, with few other major concerns due to adequate reporting of quality procedures [39,41]. The studies with the highest risk of bias were Ozen (2013) [43] and Norouzi et al., (2015) [42]. These were small RCTs with biases across all domains. All studies are reported with combined outcomes because there was no clear difference in the validity or sensitivity of individual outcome measures used in each study.

### 3.3. Outcomes of MBCT Interventions

The main findings of the seven studies are summarized in Table 1. The between-group effect sizes of MBCT for outcomes were reported for three out of seven studies and are presented in Table 2.

#### 3.3.1. Carer Stress

There was a statistically significant difference in self-rated carer stress in three studies using MBCT compared to active control groups [39,40,41]. The Kor et al., (2020) [41] study also showed a large significant reduction in BPSD related caregiver distress in the MBCT group compared to the control (Cohen’s *d* = 0.7) at six months and had the longest follow up period of 6 months. The mean PSS score at baseline of 31.8 reduced to 25.0 which is below the cut-off for high perceived stress. The Cheung et al., (2020) [39] study showed that the MBCT group had a significant improvement in stress from baseline to post intervention (PSS total score mean difference = 3.2, *SE* = 1.1, *p* = 0.03). Of significance, this [39] study compared MBCT to MBSR and showed that MBCT was better than MBSR for stress reduction in family carers (Cohen’s *d* = 0.6, *p* = 0.019). The MBCT intervention was shown to decrease self-rated carer stress compared to the pragmatic control group in the Oken et al. (2010) [35] study (but not compared to the active control). Pre–post results for stress were not significant in the tele-MBCT group of the Zarei (2018) [44] study.

The effect sizes for carer stress measured by the PSS ranged from Cohen’s *d* = 0.0 [35] to 0.4 [40] at post intervention, and increased up to 0.7 at six months in Kor et al., (2020) [41].

#### 3.3.2. Carer Burden

Carer burden was significantly reduced in the Kor et al., (2019) [40] study in the MBCT group compared to the active control at three month follow up (ZBI mean difference = −2.7, *p* = 0.006, Cohen’s *d* = 1.0). The Cheung et al., (2020) [39] study also showed within group reductions in carer burden for the MBCT group between post intervention and at three months (ZBI mean difference = 5.2, *SE* = 1.7, *p* = 0.14). Carer burden was also reduced in the Norouzi et al., (2015) [42] study, but this was only a within-MBCT group finding at two month follow up.

#### 3.3.3. Depression

There was a significant reduction in depression scores for the Kor et al. studies (2019, 2020) [40,41] in the MBCT group compared to active controls. There was a large effect size of Cohen’s *d* = 1.4 for depressive symptoms in Kor et al., (2020) [41] at six months. Within group findings for the MBCT group in the Cheung et al., [39] study also showed benefits in depressive symptoms at three months and in the Norouzi et al., (2015) [42] study at two month follow up.

The effect sizes for depression measured by the CES-D ranged from 0.04 [40] to 0.9 [41] post intervention, and increased up to 1.4 at six months [41].

#### 3.3.4. Resilience

Resilience was measured only in the two [40,41] studies and no significant differences were noted between groups in those studies.

#### 3.3.5. Quality of Life

Physical health related QOL did not change in Kor et al., (2020) [41]. However, mental health related QOL showed significant greater improvement at six months in this study with a medium effect size of Cohen’s *d* = 0.6 at six months.

#### 3.3.6. Trait Mindfulness

A statistically significant increase in mindfulness as measured by the FFMQ was found in the MBCT group at three (mean difference = 18.5, *p* < 0.01) and six months (mean difference = 19.9, *p* = 0.4) in the Kor et al., (2020) study [41]. The Cheung et al. (2020) [39] study also showed a statistically significant increase in trait mindfulness in the MBCT group at both post intervention (Helmert’s contrast mean difference = 2.4, *SE* = 1.2) and at follow up at three months (Helmert’s contrast mean difference = 2.5, *SE* = 1.2). The level of mindfulness in Kor et al., (2020) [41] was significantly correlated with improvements in a number of psychological outcomes (stress, depression, anxiety).

#### 3.3.7. BPSD in PLWD

BPSD in PLWD was not measured in most studies. The only positive result was small at three months in Kor et al., (2020) [41] (Cohen’s *d* = 0.2), but was not significant at six months.

#### 3.3.8. Adverse Effects

Only two studies [39,41] looked for any potential adverse effects or evidence of harm, and none were found.

## 4. Discussion

This systematic review showed that MBCT had beneficial effects on stress and depression for family carers of PLWD in four out of seven studies. The key finding is the large effect size for carer stress and depression in two of the studies [35,41], with results maintained at six-month follow-up in one study [41]. Our findings are similar to a previous review on MBIs in general (which were mostly MBSRs) [33]; however, results were not maintained at longer term follow up elsewhere. We found quality was an issue for the majority of MBCT studies because they were mostly small pilot RCTs with likely limitations on funding in a range of countries. The risk of bias assessment highlights the need for some objective measures by blinded outcome assessors (for example, physiological markers that are sensitive to change or clinician assessed rating scales).

### 4.1. Carer Stress

Self-perceived stress was seen as a primary outcome measure in most studies and appears valid because it is most likely to be sensitive to a mind–body intervention [35]. Of note, a large effect size for stress reduction (Cohen’s *d* = 0.7) for MBCT at six months follow up was seen in Kor et al., (2020) [41]. This is larger than studies using mindfulness interventions without a CBT component [61]. The duration of follow-up shows the potential for significant enduring stress-reduction effects of MBCT for this population, long after the intervention has ended. The mean PSS score at baseline of 31.8 suggested that most participants were experiencing high levels of stress. The reduction to 25.0 is below the cut-off for high perceived stress, suggesting this is not just statistically significant, but also clinically significant.

### 4.2. Depression and Anxiety

The large effect sizes for depression and anxiety in the large study that was adequately powered [41] are not surprising given MBCT’s original indication for recurrent depression. These results are consistent with other recent studies [62,63] and reinforce studies that show MBCT’s equivalence to antidepressant medication [26,64]. This is of significant practical implication to family carers of PLWD who have high rates of depression and anxiety [5]. Sample size was an issue for all other studies in this review and therefore they were likely underpowered to detect results of interest.

### 4.3. BPSD in PLWD

Improvements in BPSD with MBCT was noted in the Kor et al., (2020) [41] study as an immediate post intervention effect. It has been hypothesized that the calmer interactions and improvements in carer energy and wellbeing may have indirectly been of benefit to PLWD [41]. Communication with PLWD is a key component of BPSD management and because the emphasis on non-judgmental acceptance of already existent BPSD, MBCT would be of benefit to care relationships. Thus, the benefits of MBCT extend indirectly but are of potentially great significance to QOL for PLWD. It is expected that improvements in carer symptoms will translate to improvements for PLWD, and therefore BPSD is an important outcome to measure.

### 4.4. Adaptations of the MBCT Protocol

We found almost all included studies adapted the MBCT protocol, which could be helpful to enhance adherence for time-poor family carers by shortening the duration and number of sessions, and tailoring the content for carer stress rather than depression. A recent systematic review on MBIs for family carers of PLWD recommended these adaptations due to concerns that studies using the original MBSR protocol (including a 7.5 h retreat day) were thought to be associated with higher attrition rates of 10–17% [33]. These modifications were specifically made to reduce attrition rate, whilst still resulting in significant increases in trait mindfulness in the Cheung et al., (2020) [39] study. This has also been noted in other research [65] and supports the adaptation of reducing session duration and total number by at least one without losing potential active ingredients. Adaptations of the MBCT protocol do, however, make it more challenging to compare studies as they varied and adaptations were not always described in detail.

### 4.5. Skill Maintenance

Home practice is considered an essential component of the MBCT program to reinforce learnt skills that can be used for ongoing management of negative experiences in participants’ lives [66]. In the Kor et al., (2020) [41] study, the duration of home practice significantly correlated with mindfulness levels. This has been noted in previous literature [67]. One of the mechanisms by which studies sought to increase their effects may have been through extension of the original program from 8–10 weeks (by spacing out the reduced number of sessions) which increased the total time for home practice to enhance longer term maintenance of skills [41]. The long-term maintenance of skills in a self-sustaining manner is what potentially sets apart mindfulness-based interventions such as MBCT. The Cheung et al., (2020) [39] study also spaced out their protocol even further, to monthly sessions for the last three, but they noted participant feedback suggested that monthly gaps were too long.

### 4.6. The Superiority of MBCT over MBSR in This Context

The Cheung et al., (2020) [39] study that compared MBCT to MBSR gives some definitive evidence of the superiority of MBCT specifically in this population for stress reduction. Whilst the two interventions share many commonalities including structure of program, and were clearly both feasible for the population, the specific use of CBT techniques could be the key difference. Even though the family carer population is considered non-clinical, the prevalence of depressive and anxiety symptoms is sufficiently high to make CBT techniques an important beneficial component of the MBCT intervention. The MBCT protocol (unlike the MBSR protocol) also focuses on depression-specific phenomena such as negative thinking, rumination, and the consequences of low mood, and these may also have been key mechanisms to explain the stress reduction difference between the two programs for family carers. MBCT has shown superiority over MBSR in another study that compared them both, in addition to an inactive control, for patients with cardiovascular disease and comorbid depression [68].

### 4.7. Adverse Effect Reporting

There was no report of significant adverse effects in included studies which is consistent with the view that MBIs are relatively safe. However, there are known reports about harm with mindfulness meditation [69] which make it important for prospective RCTs to continue to assess for this.

### 4.8. Future Research Implications

A number of areas for future research have been identified by the authors of the included studies, and from the process of reviewing the included studies. The need for larger studies is clear, given the majority of included studies being small and of feasibility level only.

There needs to be more men included in future studies and more ethnic diversity in samples. The largest number of participants in the studies reviewed were from Hong Kong, where authors thought that the traditional Chinese population would take easily to meditation [41].

The high degree of outcome measure bias can be mitigated by the use of more objective measures such as clinician rating scales and biomarkers for stress.

The only tele-MBCT study included in this review did not show a significant reduction in carer stress (Zarei, 2018) [44]. In a post-COVID-19 pandemic world, tele-MBCT is particularly appealing for a number of reasons, but the convenience needs to be weighed against the efficacy of this modality and the equity issues faced by family carers who may not have access to high-speed internet, digital devices, or skills to use such technology.

## 5. Conclusions

In conclusion, MBCT appears to be a potentially effective intervention to reduce carer stress and improve other outcomes for family carers of PLWD. The effects seem to be sustainable with potential to also benefit PLWD. It can be delivered at low cost in relatively large groups. This has potentially significant implications on easing the public health burden of dementia internationally. Modifications of the MBCT protocol seem potentially beneficial to improve attrition rates in studies. Methodological issues noted could be used to inform future intervention studies. Large, high quality RCTs in ethnically diverse populations are required to evaluate its effectiveness for countries that are multicultural. Cost-effective larger scale health delivery also needs to be explored.

## Figures and Tables

**Figure 1 ijerph-19-00614-f001:**
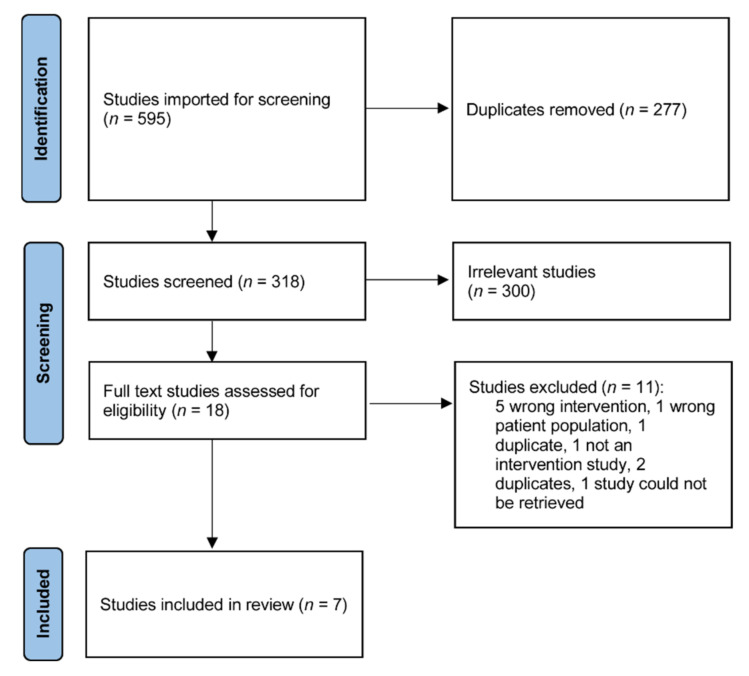
PRISMA flow chart.

**Figure 2 ijerph-19-00614-f002:**
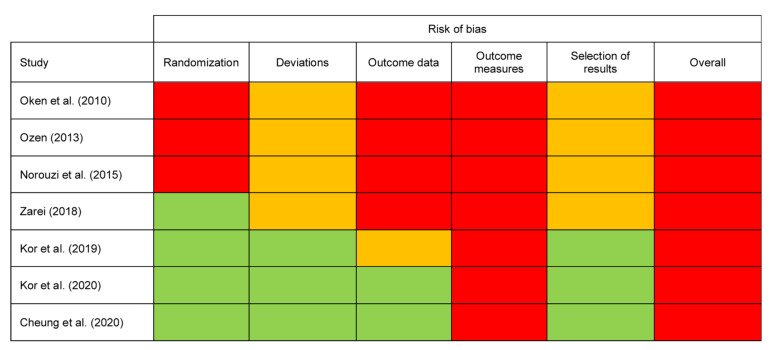
Risk of bias assessment. Note. Green = low risk, Amber = some concerns, Red = high risk.

**Table 1 ijerph-19-00614-t001:** Summary of study characteristics.

Author (Year) Country	Design	Sample	Demographics	Intervention Protocol	InterventionSelf-Practice	Control	Measures	Main Findings	Attrition Rate (%)
Oken et al., (2010) USA	Pilot single blinded 3-arm RCT with 2 controls (active A and pragmatic B).	Family carers of PLWD providing at least 12 h per week of care (*N* = 31).	Female 80.6% Mean age 64.5years (*SD* = 9.3) Care recipient relationship: Spouse 74.2% Ethnicity: Caucasian 90.3% African American 3.2% Asian 6.5%	Modified MBCT (*n* = 10) One education session on dementia weekly in-person group 90 min MBCT session for 6 weeks. Based on both MBCT and MBSR. Contents: Didactic instruction and discussion of key topics Formal meditation practices Group discussion regarding experiences and strategies for informal practice specific to time poor carers Adapted 3MBS Action plans.	Strongly encouraged to do regular daily practice with logbook records. Provided written material and audio instructions.	A. 7-week group -based education program for carers (*n* = 11). Matched to MBCT group for social support discussion time, action plan and homework. Based on Powerful Tools for Carers program with book provided for carers B. 3 h respite care once a week for 7 weeks (*n* = 10)	RMBPC reaction RMBPC confidence PSS CESD SF-36 Fatigue MAAS FFNJ GPSE PSQI ESS NPI CA CRI approach CRI avoidance Cortisol IL-6 TNF-alpha hsCRP Stroop interference ANT conflict ANT alerting Word list Expectancy Credibility	Both active interventions (MBCT and education) showed decreased self-rated carer stress compared to the respite only control. No significant difference between active groups.	12.9
Ozen et al., (2013) Canada	Pilot unblinded crossover RCT using dyads of spouses and PLWD, or spouses alone. Unpublished data.	Spouses of PLWD (*N* = 12)	Female 78% Mean Age 68.9y (*SD* = 11) Education 13.8y (*SD* = 2.2)	Modified MBCT (*n* = 12) Weekly 2 h sessions for 8 weeks Content: Formal meditation practice Informal practice Group discussion Inquiry.	Daily practice assigned as homework with self-report of time and observations. Guided meditation CDs provided.	Wait list	GDS DASS AES FFMQ QOL-AD SCS ZBI (short version) Brief COPEInventory	MBCT did not have an effect on the outcome variables examined.	25
Norouzi et al. (2015) Iran	Unblinded RCT	Female carers of PLWD with depression, low quality of life (*N* = 20)	Female 100%	Unmodified MBCT (*n* = 10) Weekly 2.5 h sessions for 8 weeks Content: Theory Practices Evaluation of tasks Assignments Group discussion	No details	Wait list	HAM-D SF-36 CBI	Reductions in depression and carer burden reported at 2 month follow up compared to their baseline within MBCT group.	0%
Zarei et al. (2018) Canada	Unblinded RCT With mixed methods. Unpublished data.	Family carers of PLWD with internet access, computer literacy, and baseline stress (*N* = 26)	Female 88% Mean age 60y (*SD* = 13) Care recipient relationship spouse 30% Tertiary educated 88.5% Employed 35% Duration of care relationship 5.12y (*SD* = 2.88) Living with PLWD 46% PLWD having Alzheimer’s disease 46%	Modified self-help and tele-MBCT (*n* = 14) Weekly 2 h sessions for 8 weeks 4–6 participants per group All received additional workbook Content: Mindfulness concepts Formal practice Modified mindful walking and movement for carer safety 2 sessions allowed to be missed and the workbook was used for these sessions Adaptation of content included carer identity and ambiguous loss as issues	Instructed to practice one exercise during the week with recording in practice log for 30–45 min per day. CDs and further readings provided in Mindful Way workbook.	Usual care	PSS SCS CES-D STAI-S CISS-SF NPI-Q Satisfaction questionnaire including 6 open questions for qualitative data.	High satisfaction with MBCT. Pre–post results in stress, depression, and anxiety were not significant in the intervention group.	8
Kor et al., (2019) Hong Kong	Pilot single blinded RCT	Family carers of PLWD providing care for at least 3 months (*N* = 36)	Female 83.3% M age 57.1 y (*SD* = 10.6) Care recipient spouse 16.7% Tertiary educated 44.4% Employed 50% Duration of care relationship 75.1 months (*SD* = 78.8) Duration of care per week 76.9 h (*SD* = 62.6)	Modified MBCT (*n* = 18) 7 sessions of 2 h each (the last 3 sessions were extended to 2 weeks apart with phone contact in between those) over 10 weeks. Single large group of 18 Content: 4th and 5th sessions of MBCT combined into one session.	Daily practice encouraged with MP3 recordings provided. Home practice duration was recorded. Weekly phone support between sessions 5 and 7	Brief education with same number of sessions and duration as intervention group. Included group sharing in addition to didactic and skills based training	PSS CESD ZBI BRS SF12_PCS SF12_MCS FFMQ HADS	The intervention group had significantly greater improvements than control for perceived stress and depression from baseline to post intervention and 3 month follow up. They also had very statistically significant reduction in burden compared to controls at the 3 month follow up.	11.1
Kor et al., (2020) Hong Kong	Multi centre Single blinded parallel group RCT 6 month follow up	Cantonese speaking family carers of PLWD providing at least 4 h of daily contact. Baseline measures suggest higher than average stress levels and lower mental health-related quality of life compared to the Hong Kong population (*N* = 113)	Female 61.1% Mean age 61.7y (*SD* = 10.5) Care recipient spouse 34.5% Duration of care relationship 71.0 months (*SD* = 91.7) Living with PLWD 69.9% Assistance from non-family 39.8% Diagnosed with more than one chronic disease 28.3%	Modified MBCT (*n* = 56) 7 sessions of 2 h each (the last 3 sessions were extended to 2 weeks apart with phone contact in between those) over 10 weeks. 3 large groups group of 17–19. Content: 4th and 5th sessions of MBCT combined into one session Psychoeducation on stress Formal practice Peer sharing Depressive relapse content replaced with information and skills for dementia caregiving Incorporating teaching on mindfulness with caregiving tasks Mindful communication with PLWD Responding to negative moods resulting from caregiving mindfully Identifying habitual emotional reactions to difficulties in caregiving.	Encouraged, documented, and monitored including during follow up by WhatsApp and emails.	Brief education and usual care with same number of sessions as intervention group. Included group sharing in addition to didactic training. Usual family care services as provided by district elderly community centres.	PSSCESD HADS (Anxiety) ZBI BRS SF12_PCS SF12_MCS NPIQ (Severity) NIPQ (Distress) FFMQ	The intervention group had greater improvement in stress, depression, anxiety, and BPSD-related caregiver distress, compared to control at both post intervention and 6 month follow up.	7
Cheung et al., (2020) Hong Kong	Single blinded RCT.	Family carers of PLWD, providing care for at least 3 months (*N* = 53)	Female 86.8% Care recipient spouse 7.5% Ethnicity All Chinese Mean duration of providing care to PLWD 8.71 years (*SD* = 10.56)	Modified MBCT (*n* = 26) 7 sessions of 2.5 h over 16 weeks (first 4 sessions were weekly then the last 3 were monthly with phone contact in between) Content: Focus on addressing low moods and negative thoughts to help participants gain experience in recognising emotional symptoms and gain confidence early.	CD recording of all exercises provided.	Modified MBSR. Same number of sessions, duration, and frequency (including phone support) as MBCT group. Delivered by same therapist as MBCT. Adaptations made by same panel of expert clinicians as MBCT. Fidelity checking done.	PSS CESD ZBI FFMQ-SF	Both interventions were feasible. Both groups had positive within-group effects on perceived stress, depression and burden, while the MBCT group had a larger effect on stress reduction than the MBSR group.	3.8

Notes. 3MBS = 3 min breathing space; AES = Apathy Evaluation Scale *(informant version)*; ANT = Attentional Network Test; BRS = Brief Resilience Scale; CA = Caregiver Appraisal; CBI = Caregiver Burden Inventory; CES-D = Centre for Epidemiological Studies-Depression Scale; CISS-SF = Coping Inventory in Stressful Situation-Short Form; CRI = Coping Responses Inventory; DASS = Depression Anxiety Stress Scale; ESS = Epworth Sleepiness Scale; FFMQ = Five-Facet Mindfulness Questionnaire; FFMQ-FS = Five-Facet Mindfulness Questionnaire Short Form; FFNJ = Measure of being non-judgemental adapted from factor five; GDS = Geriatric Depression Scale; GPSE = General Perceived Self-Efficacy; HADS = Hospital Anxiety and Depression Scale; HAM-D = Hamilton Depression Rating Scale; hsCRP = High Sensitivity C-Reactive Protein; IL-6 = interleukin-6; MAAS = The Mindful Attention Awareness Scale; NPI = Neuropsychiatric Inventory; RMBPC = Revised Memory and Behaviour Problems Checklist; PSQI = Pittsburgh Sleep Quality Index; PSS = Perceived Stress Scale; QOL-AD = Quality of Life in Alzheimer’s Disease (*informant version*); SCS = Self-Compassion Scale; SF12-PCS = Short Form 12 Physical Component Summary Score; SF12-MCS = Short From 12 Mental Component Summary Score; SF-36 = Medical Outcomes Study Short-Form Health Survey; STAI-S = State-Trait Anxiety Inventory—Short Version; TNF = alpha Tumour Necrosis Factor–alpha; ZBI = Zarit Burden Interview.

**Table 2 ijerph-19-00614-t002:** Effect sizes of included studies.

Study	Outcome Measure	Effect Size (*d*)		
		Post Intervention	3 Months Post Intervention	6 Months Post Intervention
Oken et al., (2010)	PSS	0.0		
	CES-D	0.3		
Kor et al., (2020)	PSS	0.4		0.7
	CES-D	0.9		1.4
	HADS (Anxiety)	0.7		1.0
	ZBI	0.7		0.6
	BRS	0.1		0.3
	SF12-PCS	0.5		0.04
	SF12-MCS	0.1		0.6
	NPIQ (Severity)	0.2		0.3
	NPIQ (Distress)	0.4		0.8
Kor et al., (2019)	PSS	0.4	0.2	
	CES-D	0.04	0.77	
	HADS (Anxiety)	0.35	0.08	
	ZBI	0.71	0.13	
	BRS	0.64	0.16	
	SF12-PCS	0.24	0.24	
	SF12-MCS	0.17	0.17	

Note. BRS = Brief Resilience Scale; CES-D = Center for Epidemiological Studies–Depression Scale; HADS = Hospital Anxiety and Depression Scale; NPI = Neuropsychiatric Inventory; PSS = Perceived Stress Scale; SF12-PCS = Short Form 12 Physical Component Summary Score; SF12-MCS = Short From 12 Mental Component Summary Score; ZBI = Zarit Burden Interview.

## Data Availability

Not applicable.

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
