# Peer review of "Mindfulness-Based Cognitive Therapy for Stress Reduction in Family Carers of People Living with Dementia: A Systematic Review"

_ijerph, 2022, doi:10.3390/ijerph19010614_

Round 1

Reviewer 1 Report

Dear Authors,

The work is very thorough. The research though complex, has tried to avert all possible biases. In particular, proper attention has been paid to selection bias. Only aspect that I suggest to revise is the paragraph of "Future Research Implication". Rather than the future implications, the authors focus on the limits of research, which is right after all. Authors should describe the implications more, as research has an important application potential.

I therefore recommend this single minor review.

Best regards!

Author Response

Dear Reviewer,

Thankyou for your kind comments and review of our paper.

With regards to your revision request for the section titled "future implications of research", we agree that the focus of this section has been to note the limitations of the included studies that suggest areas that need further research (eg sample size, gender and ethnic diversity). We have also noted the significance of the on-line modality particularly in our pandemic times currently but tensions associated with online methods of delivery. 

In terms of requesting us to describe the implications more, can we please direct you to the section in the conclusion as below which specifically addresses this. In order to not be repetitive in the manuscript, we hope that the below section adequately addresses the importance of the application of this research:

"MBCT appears to be a potentially effective intervention to reduce carer stress and improve other outcomes for family carers of PLWD. The effects seem to be sustainable with potential to also benefit PLWD. It can be delivered at low cost in relatively large groups. This has potentially significant implications on easing the public health burden of dementia internationally. Modifications of the MBCT protocol seem potentially beneficial to improve attrition rates in studies. Methodological issues noted could be used to inform future intervention studies. Large, high quality RCTs in ethnically diverse populations are required to evaluate its effectiveness for countries that are multicultural. Cost-effective larger scale health delivery also needs to be explored."

We have also added further practical implications into the discussion section as this was requested by another reviewer as below. We hope that all together these revisions are now satisfactory to you wanting further details on important implications:

We have highlighted the significant practical implication of long term benefits of MBCT for stress reduction in the discussion section's new subsection 4.1 carer stress 

"The duration of follow-up shows the potential for significant enduring stress-reduction effects of MBCT for this population, long after the intervention has ended."

We have also highlighted the practical implication of large effect sizes for anxiety and depression for this vulnerable population in  the discussion section's new subsection 4.2 depression and anxiety

"This is of significant practical implication to family carers of PLWD who have high rates of depression and anxiety [5]."

We have already mentioned the potential practical implications for PLWD in the discussion section as well.

thank you.

Reviewer 2 Report

Thank you for the opportunity to review this manuscript about Mindfulness-Based Cognitive Therapy for family carers. Such an important topic. The review is carefully conducted and reported. Please find below some comments for authors.

Methods

  • I suggest authors to spesify the reporting checklist used, PRISMA perhaps?
  • I also suggest to include a full search strategy to at least one database. I was also wondering did you use family as a search term?

Results

- I was wondering about the ethnicity of the participants, while 3 of the studies are from Hong Kong. Are all the participants Chinese? In the current text, only Cheung is mentioned.

Discussion

- Maybe some practical implications could be given as well

Table 1

- Intervention protocol and measures are a bit difficult to read while text begins from different places.

Author Response

Dear Reviewer,

Thankyou for your comments and appreciation of the importance of this review topic. 

Please find our revisions as below:

  1. Methods: We have used the PRISMA 2009 reporting checklist and now include this in our manuscript in the materials and methods section (section 2, page 3, line 117)
  2. Methods: We have included a full database search strategy as supplementary information and indicated that this is available in the main text under search strategy (2.1) for the methods section on page 3. Yes we used family and many variants of this term in our search terms (please see line 6 of the search strategy). 
  3. Results: Ethnicity data was only reported in 2 studies (Oken et al, 2010 and Cheung et al. 2020). The other two studies that were conducted in Hong Kong unfortunately did not report ethnicity data so we were not able to assume that these studies were all Chinese. Therefore we have not commented on these two studies. 
  4. Discussion: We have now added the significant practical implication of long term benefits of MBCT for stress reduction in the discussion section page 15, line 23,24,25:  "The duration of follow-up shows the potential for significant enduring stress-reduction effects of MBCT for this population, long after the intervention has ended." We have also now further highlighted the practical implication of large effect sizes for anxiety and depression for this vulnerable population given the high rates of these disorders for them.  "This is of significant practical implication to family carers of PLWD who have high rates of depression and anxiety [5]." We have already mentioned the potential practical implications for PLWD indirectly in the discussion section.
  5. Table 1: We have now centred left the text in the table, and hope the journal can assist further with formatting to make this more presentable if the manuscript is accepted for publication.

thank you.

Reviewer 3 Report

The subject considered is an interesting one. The logic of the Introduction is good, the selection criteria (2.2 on page 3) are clear. However, authors should improve the logic of section Result and section Discussion.

Section 3.1.4, page 6:
Although the first paragraph mentioned eight or more indicators, the authors only reported two clearly and simply listed the rest in one paragraph. 

Section 4, page 15:
The logic and links between paragraphs are not strong or clear enough. It's more like a summary of pieces of literature reviewed in the present study but not a discussion. I suggest authors discuss more research implications.

Author Response

Dear reviewer,

thank you for your review of our paper. We agree that there can be better structuring overall, and this is consistent with other reviewer's feedback. Therefore we have added many subheadings to the paper. The introduction section has now got 5 subheadings. The Risk of Bias section has been completely revised to structure the content within the domains covered in the risk of bias tool used. The results section (3.3) has also been re-structured with subheadings to cover all the outcomes. The discussion section has also been re-structured to make it more readable with relevant subheadings. We hope these substantial changes will help with readability. Your specific concerns raised have also been addressed as follows:

1. We have restructured section 3.1.4. so that it all outcome measures are described in separate paragraphs now.

2. We have completely restructured discussion section 4. to now include sub-headings as recommended by another reviewer. We hope this helps the readability of this section now. We agree that there needs to be more emphasis on practical implications. The importance of enduring benefits of MBCT for this population have been highlighted. The significance of large effect sizes on depression have been highlighted. The need to mitigate bias (especially outcome measure bias which is an issue for all studies reported) and how this may be done is also included. The following statements have been added in the discussion section now:

"The duration of follow-up shows the potential for significant enduring stress-reduction effects of MBCT for this population, long after the intervention has ended." page 15 line 21 "This is of significant practical implication to family carers of PLWD who have high rates of depression and anxiety [5]". page 15, line 34,35

"The high degree of outcome measure bias can be mitigated by the use of more objective measures like clinician rating scales and biomarkers for stress." page 17, line 107, 108

Reviewer 4 Report

Manuscript ID: ijerph-1452229 

Rationale: The study adopts a solid research methodology (systematic review) and includes all necessary information regarding this review process. The research goals are clearly stated and evaluated through a synthesis and analysis of the results of 7 intervention studies which included control groups. The results are interpreted in the context of the independent variable (mindfulness-based cognitive therapy), with some clear implications for future research and practice. Overall, the paper is well-written. I have a few suggestions regarding formatting, but believe that the paper is of interest to readers and provides a clear contribution.

Overview: This study adopted a systematic review to evaluate the potential effectiveness of mindfulness-based cognitive therapy (MBCT) in carers of individuals with dementia, following standard PRISMA guidelines. This study addresses a research gap regarding the use of MBCT for carers of individuals with dementia, as developed by a clear review of the literature. The search strategy, eligibility criteria, and bias assessment were detailed, as was the process of screening articles for inclusion. Although the final number of 7 articles was fairly small, definite trends were found in terms of the effectiveness of MBCT in terms of the outcomes of carer stress, burden, depression, resilience, and other psychological factors. The results were interpreted in terms of the outcome variables and explained by reference to elements of MBCT which may make this approach more suitable and effective as compared to other interventions. Implications for future research are provided, including the use of tele-MBCT in a post-COVID-19 world, and conclusion are provided which adequately summarize the main findings of the study.

My comments to the authors are included below:

Dear authors,

It was a pleasure to read your manuscript. Although the topic of mindfulness-based interventions is receiving a great deal of attention, your emphasis on the effects of MBCT on carers of PLWD offers a new perspective and clear insights and implications. Below I will include an overview of the article, followed by some recommendations.

Overview: This study adopted a systematic review to evaluate the potential effectiveness of mindfulness-based cognitive therapy (MBCT) in carers of individuals with dementia, following standard PRISMA guidelines. This study addresses a research gap regarding the use of MBCT for carers of individuals with dementia, as developed by a clear review of the literature. The search strategy, eligibility criteria, and bias assessment were detailed, as was the process of screening articles for inclusion. Although the final number of 7 articles was fairly small, definite trends were found in terms of the effectiveness of MBCT in terms of the outcomes of carer stress, burden, depression, resilience, and other psychological factors. The results were interpreted in terms of the outcome variables and explained by reference to elements of MBCT which may make this approach more suitable and effective as compared to other interventions. Implications for future research are provided, including the use of tele-MBCT in a post-COVID-19 world, and conclusion are provided which adequately summarize the main findings of the study.

Suggestions:

  1. Organization: I suggest the use of sub-headings for certain sections of the manuscript. For the Introduction, you can include sub-headings such as "psychological factors related to carers of PLWD," "traditional stress-reduction interventions," "mindfulness-based interventions," and "research gap/rational" in addition to your sub-heading of "aims." For Section 3.2, please consider organizing the content according to the risk of bias assessment categories as stated in Figure 2. For Section 3.3, as this section covers the results of individual studies, consider a different heading, such as "Outcomes of MBCT Interventions" (as opposed to "Results of Individual Studies" since this suggests you will cover each study one at a time). Then, you might include sub-headings for the main outcomes (as they relate to your research aims). A similar organization of the Discussion section using outcome variables for sub-headings is suggested.
  2. Description of methodology: While it is claimed that a narrative synthesis approach was adopted, the manuscript does not follow the style and format of other articles of this nature. Perhaps "systematic review" is a meaningful enough description of your study, without the need to get into the details required for a "narrative synthesis." If you are sure that this is a "narrative synthesis" and meets all relevant criteria, then a citation and more detailed explanation are required in section 2.5.
  3. Table 1: I suggest that a simplified version be provided in the manuscript (with author/date, design, sample, and main findings) which can fit on one page. Then, the entire table, in its current form, could be added as an appendix. Also, please check the formatting of the content to ensure that it is left aligned rather than centered for the sake of appearance.
  4. Interpretation of post-intervention effect sizes: The findings by Oken et al., 2010 and Kor et al., 2020 of a significant increase in the effect size after 6 months could be interpreted more clearly and stated more strongly as a key finding in the Discussion section.
  5. Limitations: The PRISMA 2020 checklist indicates that limitations should be stated clearly. One limitation may be the sample size. Others, such as the high degree of potential bias, should also be more clearly mentioned with suggestions for overcoming these shortcomings in future research.
  6. Additional keywords: cognitive therapy, depression, anxiety, systematic review are other options.
  7. Formatting of references: The style should follow the journal's guidelines (rather than APA).
  8. PRISMA 2020: Was the checklist used and followed? If so, this could be mentioned and cited.

Author Response

Dear reviewer, Thankyou for your kind words, and your comprehensive review of our paper. We have made changes to almost all areas you suggest as per below:   1. We agree, more subheadings will be useful for the reader to better navigate through the text. We have added all your recommended ones to introduction section. We have re-arranged the risk of bias section to use subheadings that cover the domains of risk covered. We have used subheadings in the outcomes section consistent with aims of the review, and finally have also added subheadings in the discussion section as recommended.   

2. The term "narrative synthesis" has been removed from the abstract and the methods section, and the following statement has been modified accordingly:   "a systematic review approach was planned with information presented in text and tables to summarize and explain the characteristics and outcomes of included studies. Findings both within and between included studies were explored."  

3. We would like to keep table 1 with its current contents as this is typical of other systematic reviews. However, the formatting has been improved for readability and we hope the journal can assist further with this to shorten the number of pages that this table takes up.   

4. We agree, it is helpful to make the key findings of this review more clear in the discussion section. Therefore we have added the statement below about the effect sizes early in the discussion. (As part of the re-organisation of this discussion section, we hope it is easier to also now see the details of the effect sizes in each relevant subsection further below).   "The key finding is the large effect size for carer stress and depression in two of the studies [35,40], with results maintained at six-month follow-up in one study [40]."  

5. We agree, and have added to the future research implications section, the sentence about mitigating outcome measure bias in particular. (We have already noted other areas of limitations such as sample size in this subsection of the discussion section already).

"The high degree of outcome measure bias can be mitigated by the use of more objective measures like clinician rating scales and biomarkers for stress."

6. Thankyou. Additional key words have been added:

"cognitive therapy, depression, systematic review"

7. The journal was happy for us to submit with any referencing style. We are very happy to change over to the recommended style once accepted for publication.

8. The PRISMA 2009 checklist was used and has now been cited. The PRISMA 2020 checklist was not available at the time when the search was conducted. 

Reviewer 5 Report

Thanks for reviewing this article which admitted important interventions to alleviate the major burdensome caregiver burden for people living with dementia.

One of the very popular non-pharmacological interventions for caregiving stress is mindfulness-based intervention such as mindfulness-based cognitive therapy for reducing the stress experienced by family carers or even formal caregivers. 

Despite the relatively small number of eligible studies, this review could give a potential trending of the effectiveness of the results and future usage of these techniques. 

A minor inquiry about the selected methodology " the narrative synthesis", would be more declarative to explain it more in the methodology section in contrast to the standard systematic literature review in terms of source selection, evaluation, and assessment.

Author Response

Dear Reviewer,

Thankyou for your comments and for valuing the topic of this review.

We have decided to remove the term, "narrative synthesis" entirely from the methods section of this systematic review. Another reviewer has also questioned its use, and we agree with them that the term "systematic review" alone is sufficient and a more accurate description of the method used here. 

Therefore we have changed the relevant sentence in the abstract and section 2.5 data analysis sections as below:

in abstract: "Quantitative findings were explored".  

in 2.5 data analysis:  "..a systematic review approach was planned with information presented in text and tables to summarize and explain the characteristics and outcomes of included studies. Findings both within and between included studies were explored." 

Reviewer 6 Report

The paper to be reviewed has an adequate introduction, justification of the contents to be studied as well as the results.  

In the section "materials and methods" is indicated " Experts were contacted to ensure saturation of literature.". It should be better explained how this contact was made.

When you search in "PROSPERO" for the article “A systematic review of intervention studies using mindfulness-based cognitive therapy to reduce carer stress/burden and improve quality of life in family carers of people with dementia [CRD42020186414]” to be indicated “Review Ongoing”. It has not yet been registered.

Author Response

Dear reviewer,

thank you for your comments and helpful recommendations. We have addressed the two main areas you raise as below:

  1. We had contacted by email the authors of 4 out of 7 of the studies included in this review to check if they were aware of any further studies that were relevant. We made sure to check with the most published author in the field (Kor) to ensure this was also the case at the time of finalising the search results.  We have included the following statement to explain how the contact was made in the manuscript:

'authors of four identified studies were emailed to ask about other studies that they were aware of'

2. This study was definitely registered with PROSPERO. We have attached a screenshot of the website showing that it is registered. Please note that due to the pandemic, PROSPERO automatically registered this study without eligibility checking. We also have email confirmation but can only upload one document as evidence so have chosen to attach the website screenshot as evidence for you.

Thankyou.
